# Complete genome sequencing and characterization of a potential new genotype of Citrus tristeza virus in Iran

**Abozar Ghorbani**[1]*, **Mohammad Mehdi Faghihi**[2], **Faezeh Falaki**[3], **Keramatollah Izadpanah**[4]

**1** Nuclear Science and Technology Research Institute, Nuclear Agriculture Research School, Karaj, Iran,
**2** Plant Protection Research Department, Fars Agricultural and Natural Resources Research and Education Centre, AREEO, Zarghan, Iran, **3** Department of Plant Protection, College of Agriculture Sciences and Food Industries, Science and Research Branch, Islamic Azad University, Tehran, Iran, **4** Plant Virology Research Centre, College of Agriculture, Shiraz University, Shiraz, Iran

* abghorbany@aeoi.org.ir

**Data Availability Statement:** All relevant data are within the paper and its Supporting Information files.

## Abstract

Citrus tristeza virus (CTV) is one of the economically destructive viruses affecting citrus trees worldwide, causing significant losses in fruit production. Comparative genomic studies have shown genetic diversity in various regions of the genome of CTV isolates, which has classified the virus into several genotypes. In recent years, some orange citrumelo-tolerant rootstocks showed yellowing, decline, and vein clearing in northern Iran (Mazandaran province, Sari). We confirmed the presence of CTV in the symptomatic trees by reverse transcription PCR (RT-PCR). The complete genome of a Sari isolate of CTV (Sari isolate) was sequenced using next-generation sequencing (NGS) technology. In addition, phylogenetic analysis, differential gene expression of the virus and identification of its variants in a population were studied. We obtained the final contigs of the virus (nt) and annotated all genomes to viral ORFs, untranslated regions (UTRs), intergenic regions, and 5' and 3' ends of the genome. Phylogenetic analysis of the Sari isolate and other genotypes of CTV showed that the Sari isolates were placed in a distinct cluster without a sister group. Based on the number of specific transcripts (TPM) in CTV RNA -Seq, P13 was the most highly expressed gene related to the host range of the virus and its systemic infection. The ORFs of the polyprotein, P33, and P18 showed variation in a single population of the sari isolate. The CTV has a potential for variation in a population in a host, and these variations may contribute to the best fit of the CTV in different situations. In Iran, whole genome sequencing of the CTV was performed for the first time, and we gained new insights into CTV variation in a population.

## Introduction

Citrus tristeza is an economically destructive disease affecting citrus trees throughout the world, causing significant losses in fruit production [1]. The causal agent of the disease, Citrus tristeza virus (CTV), is an RNA plant virus of the genus *Closterovirus* (family *Closteroviridae*)

**Funding:** The authors received no specific funding for this work.

**Competing interests:** The authors have declared that no competing interests exist.

composed of a variety of strains. The virus host range is limited to *Citrus* and related species [2]. The symptomology and severity of CTV infection widely vary depending on the type of viral strain, host cultivar, and root-stock/scion combination. However, classic disease symptoms include stem-pitting, vein clearing, leaf cupping, and yellowing [3]. The CTV is distributed in the world as mild or severe strains. The main mode of transmission of CTV is through multiple aphid vectors including *Aphis gossypii*, *Toxoptera citricida*, *A. citricola*, *T. aurantii*, *A. craccivora*, and *Myzus persicae* [4–6]. Other spread pathways include graft transmission and plant material infested with aphid vectors [7].

CTV is a single-stranded positive-sense RNA virus and the genome is encapsulated by the major and minor coat proteins (CP and CPm, respectively) [8]. The complete genomes have been sequenced from the biologically and geographically distinct strains and reveal that the CTV has the largest plant viral genome. The sequences range from 19,226 to 19,306 nucleotides and are organized into 12 open reading frames (ORFs) which potentially encode for 17 proteins [9–11]. The most comprehensively studied part of the CTV genome is the CP gene which encodes for the major coat protein (25 KDa). The CP gene is commonly used for molecular typing of CTV isolates around the world [9,12–14].

ORF1 encodes a replication-associated protein (p349) directly translated from the genome (gRNA) and proteins p33, p6, p65, p61, p27, p25, p18, p13, p20 and p23 (5' to 3') expressed via 3′-terminal subgenomic RNAs (sgRNAs). Based on genome-wide sequence diversity, CTV isolates are classified as strains or genotypes. The recognized genotypes of CTV are constantly expanding and to date include T36, VT, T3, RB, T68, T30, HA16-5, S1, AT -1, and Taiwan-Pum. CTV strains are known to recombine frequently and the Maximum likelihood (ML) phylogenetic method has been used for classification [15].

Factors that may influence genetic diversity include geographic differences and mode of transmission [16]. Studying the whole genome of diverse isolates from around the world provides a deeper understanding of molecular epidemiology. In particular, the use of next-generation sequencing (NGS) tools such as RNA sequencing (RNA-Seq) is now a major focus in the field of plant virology for accurate and specific whole-genome analyses and rapid diagnoses [17,18]. In this study, high-throughput sequencing (HTS) was used to determine the genetic diversity and phylogenetic relationships for the Sari isolate of CTV.

## Materials and methods

### Ethics statement

This research was carried out in the laboratories of the Shiraz University and Nuclear Science and Technology Research Institute. No other permits were required to conduct this research. We also confirm that no endangered or protected species were involved in the studies.

### Sample collection

Samples of leaves and young shoot tips of sweet orange trees on citrumelo rootstocks showing various virus-like symptoms, including stem-pitting, vein clearing, leaf cupping, and yellowing, were collected in Mazandaran Province, Sari, Iran, in 2021–22. A total of five orchards were visited, and samples were stored individually in microfuge tubes at −80˚C until processing. Samples were used for RT-PCR and RNA-Seq tests.

### RAN extraction and initial RT-PCR screening of CTV

Total RNA was extracted from 100 mg of leaves using the TRIzol® reagent (USA) following the manufacturer's instructions. RNA was quantified by Nanodrop® spectrophotometer

(Thermo Fisher Scientific, USA). Total RNA was treated by DNase (Thermo Scientific). For initial RT-PCR screening of CTV, cDNA was synthesized with Random hexamer primers and an M-MuLV Reverse Transcriptase cDNA synthesis kit (Thermo Scientific). RT-PCR reaction was carried out in a total volume of 20 μL of reaction mixture containing 1 μL of cDNA as template, Taq DNA polymerase (Takara Bio, Otsu, Shiga, Japan), (1.25 U/50 μL) and capsid protein gene primers (forward: 5′ ATGGACGACGAAACAAAGAA 3′), (reverse: 5′ TCAACGTGTG TTGAATTTCC 3′) [19]. PCR reactions were performed using the Ampliqon Taq DNA Polymerase 2x Master Mix (Denmark) in a total volume of 12.5 **μL** containing 20 ng cDNA and 10 μM of each primer. The PCR condition comprised one initial denaturation cycle at 95˚C for 10 min, followed by 35 cycles of 95˚C for 30 s, 55˚C for 30 s, and 75˚C for 30 s, with a final extension step at 75˚C for 10 min. Aliquots of PCR-amplified fragments were loaded on 1% agarose gels in Tris-borate (TBE) buffer (0.09 M Tris base, 0.09 M boric acid, 0.002 M EDTA, pH 8.0). A 100-bp DNA Ladder (Promega, Madison, WI) was used as a nucleic acid marker. After electrophoresis, the gels were stained with ethidium bromide at 0.5 μg/ml and then analyzed using a BIO imaging system (Syngene, Frederick, MD).

## RNA library prep and CTV whole genome sequencing

Total RNA from three leaf samples of a tree was used for RNA-Seq. The rRNA was removed from the total RNA using a Ribo-zero rRNA Removal Kit (Epicentre, WI, USA). RNA-Seq libraries using TruSeq Stranded Total RNA for Illumina were prepared according to the manufacturer's instructions and sequenced on the Illumina HiSeq 2000 platform (Novogene, China), generating paired-end reads of 150 bp.

## Bioinformatics analysis

Data analysis of FASTQ files was performed using CLC Genomics Workbench (version 20, QIAGEN, Venlo, The Netherlands). Sequencing adaptors and low-quality sequences with ambiguous nucleotides were trimmed to obtain sequences of approximate size (using default parameters: Reads less than 15 nt were trimmed, ambiguous nucleotides maximum 2).

FASTQ files of paired-end sequences from the library were assembled into transcriptomes de novo using CLC Genomic Workbench (word size 15, minimum length of contigs 150 nt). To distinguish and retrieve viral sequences from the entire transcriptome, the assembled transcriptome contigs were mapped to the chromosol and non-chromosol reference genome of Citrus (Citrus sinensis, GCF_000317415). Subsequently, the unmapped contigs were compared with sequences available in the NCBI viral reference database (https://www.ncbi.nlm. nih.gov/genome/viruses/) using the CLC BLAST tool, which is more reliable than other sequence similarity programs for virus identification (the E-value cut-off was 1e-5). Following the initial analysis, putative virus-associated contigs were compared with sequences in the NCBI NR (non-redundant proteins) database. Subsequently, endogenous virus-like sequences were removed from the data set, and virus-related contigs were retained for further analysis.

## Viral sequence mapping and genome assembly

To assemble the whole viral genomes, the reads transcriptome sequences were aligned with a reference whole viral genome sequence (NC_001661). Sequences associated with the viral genome were mapped to the viral reference genome using Geneious Prime 2019, and consensus sequences were generated with a threshold of 95% identity.

The viral-associated sequences were then analyzed using Geneious version R10 (Biomatters, New Zealand) for sequence trimming, nucleotide analysis, viral ORF determination, gene annotation, and phylogenetic analysis. In addition, the phylogenetic maximum likelihood tree

was constructed using a 50% bootstrap threshold with 1000 bootstrap repeats and a score defined using the Kimura 2-parameter model (MEGA 11).

## Genomic diversity of CTV at interpopulation and intrapopulation levels

To evaluate the genetic diversity of the selected CTVs at the ORF level, new reference sequences (accession number: OP900953) were generated from the RNA-Seq data examined using CLC Genomics Workbench. The clean reads were mapped to the viral contig genome. The minimum coverage, minimum variant frequency, and maximum variant P values were set to 2, 0.01, and $10^{-6}$, respectively. Single-nucleotide variants (SNVs) were filtered out as synonymous SNVs. The frequency and distribution of polymorphisms in ORFs of CTV (Sari isolate) for which transcriptome profiling was performed were assessed and values were compared. The Protein Data Bank (PDB) was downloaded from RCSB PDB (https://www.rcsb.org) to visualize SNVs in tertiary protein structures using CLC Genomics Workbench.

## Profiling the gene expression using RNA-Seq data

Reads trimmed to RNA-Seq data were subjected to expression analysis by mapping all sequences to the synthesized master sequences. Transcript per million (TPM) was calculated for the detected ORFs using CLC Genomics Workbench, with the default parameters of the software and the length fraction and similarity fraction of 0.8.

## Phylogenetic analysis

Phylogenetic trees for sequences deposited in the NCBI nucleotide database were constructed based on whole-genome and phylogenetic analysis. Briefly, the tree construction was based on the ClustalW alignment of the concatenating sequence of CTV using Geneious Prime 2019 and applying the maximum likelihood method in the MEGA11 program (Tamura et al., 2021). The substitution models used for each phylogenetic tree were selected via the best-fit model tool of MEGA software. Carrot yellow leaf virus (NC013007) was used as an out-group to root the tree, and bootstrapping (threshold: 60) was carried out using 1000 replicates.

# Results and discussion

## Symptomatology and conventional RT-PCR

During the 2021–22 field surveys in different districts of **Mazandaran Province, Sari, Iran,** sweet orange trees (on citrumelo rootstocks) showed the typical tristeza symptoms including chlorosis, yellow leaves, leaf cupping, vein clearing, vein flecking, stem pitting and grooving, poor growth and decline condition, (Fig 1). Samples that showed diverse virus-like symptoms were screened by RT-PCR for the presence of CTV. The virus was detected in all symptomatic samples (5 samples) as indicated by the amplification of specific fragments with approximately 670 bp (Fig 2). No amplification was observed with the RNA templates of healthy citrus plants.

## Genome assembly of CTV

After preprocessing the raw data (42,133,736 reads, 150 nt), clean reads (39,652,356) were obtained from the whole transcriptome sequencing sample. Subsequently, citrus-related reads (80%) were removed by comparing transcriptome contigs with the citrus sequences available in NCBI using MEGABLAST. Assembly of the remaining viral-associated reads produced 0.6% CTV-related contigs of 300 to 18,397 nt. No additional contigs with high sequence similarity to other viruses or viroids were detected from the transcriptomic reads. After assembly, we obtained the final contigs of the virus (19,300 nt), and we annotated all genomes at viral

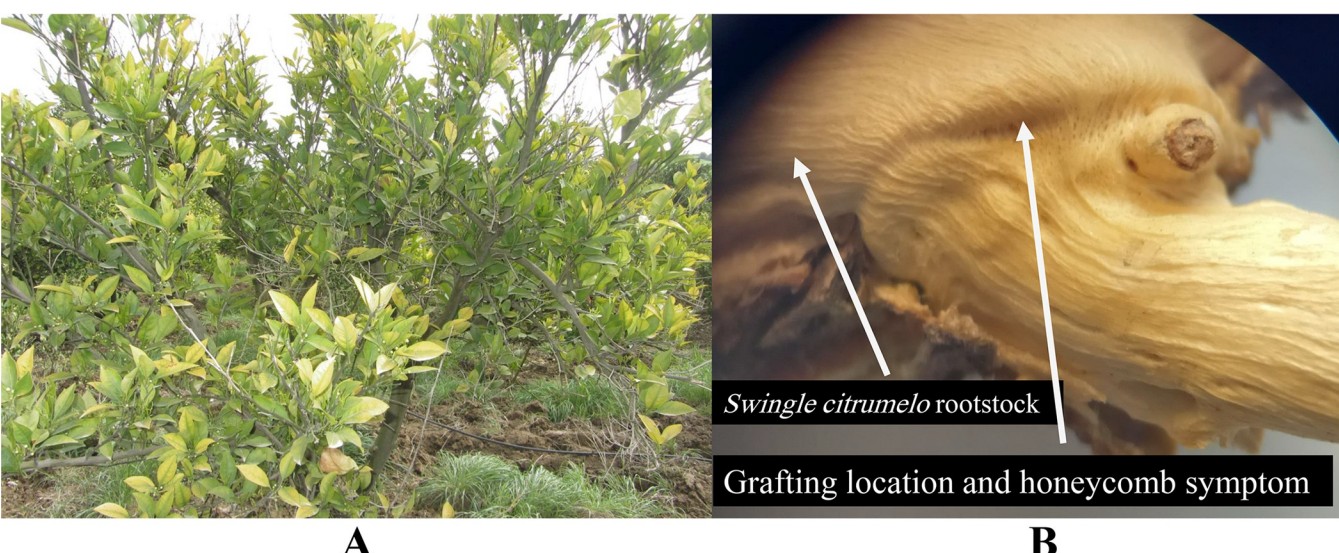

**Fig 1. Citrus tristeza virus symptoms on sweet orange trees on citrumelo rootstocks.** Yellowing of leaves (A) andstem pitting and grooving at graft union (B).

ORFs, untranslated regions (UTRs), intergenic regions, and 5' and 3' ends of the genome (Fig 3 and Table 1).

## Phylogenetic analysis of full-length CTV sequences

Phylogenetic analysis of the Sari isolate and other genotypes of CTV that were downloaded from NCBI was performed. The maximum likelihood method and whole genome sequence of

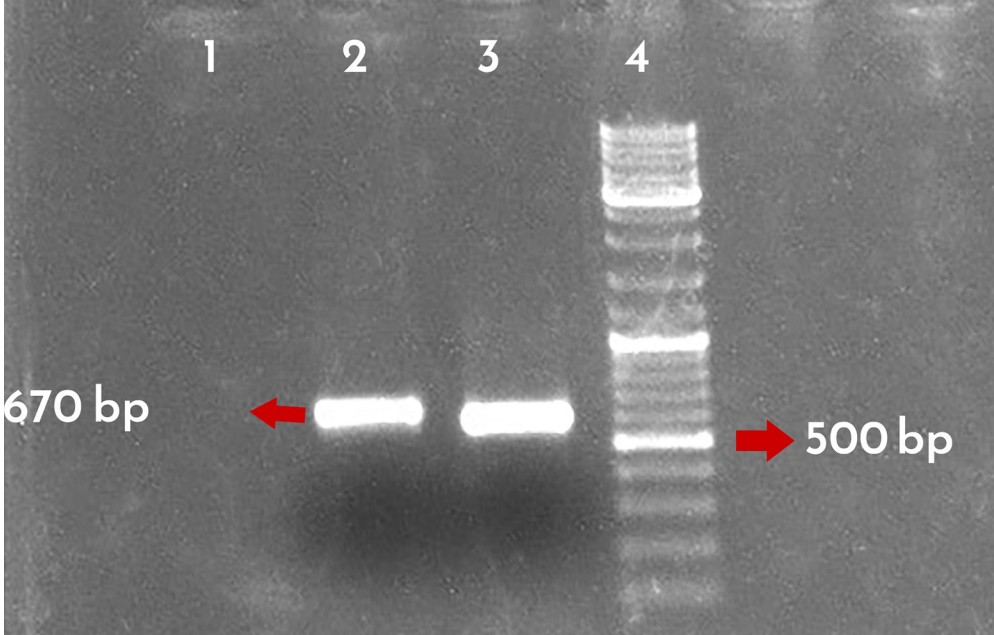

**Fig 2. Electrophoresis pattern of RT-PCR products using total RNA extracts.** Lane 1: A healthy orange seedling, Lane 2: Infected trees, Lane 3: CTV positive control, Lane 4: DNA ladder (Thermo Scientific, UK).

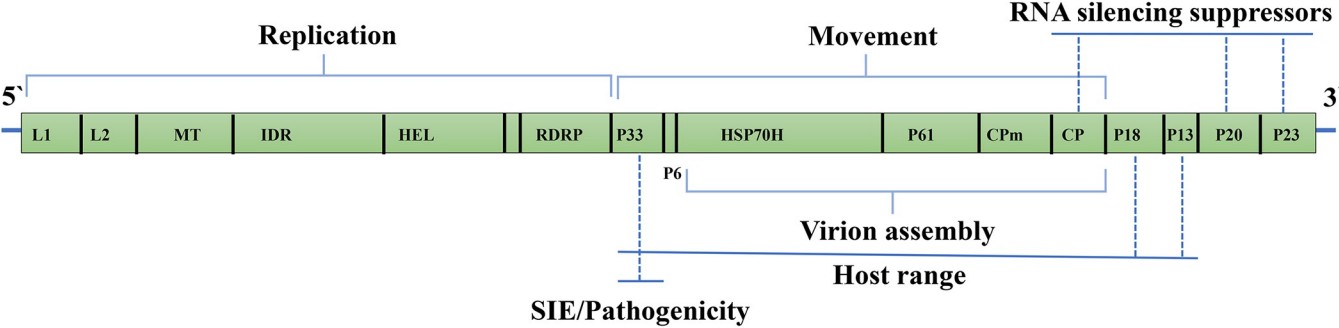

**Fig 3. Schematic diagram of the CTV genome organization.** The open boxes represent ORFs and their translation products.

CTV genotypes were used for genotype clustering. Phylogenetic analysis of our data revealed that the Sari isolate (accession number OP900953) was placed in a distinct cluster from other genotypes and not in a sister group with a known genotype (Fig 4). On the other hand, identifying the similarity matrix of the Sari isolate with others showed at least 80% identity with other genotypes and the highest similarity (92%) with the T3 genotype (S1 File).

## Differential expression of CTV ORFs

Raw transcriptomic data were used to profile the differential gene expression of CTV in citrus transcriptomic data. Differentially expressed genes (DEGs) were characterized for RNA-Seq data (adjusted-value $p < 0.01$) (Fig 5). Based on the number of specific transcripts (TPM) identified in CTV RNA-Seq. P13 which is positioned toward the 3′ termini of the genome near the CP gene, was the most highly expressed gene than the 5′ terminal gene encoding polyprotein and P6 protein which is related to the host range (Fig 5). P13 has the same function as P18

**Table 1. Genome features and ORFs ofCTV sequence of Sari isolate.**

| Type | Length (nt) | Product | Locus_tag |
|---|---|---|---|
| **3'UTR** | 273 | | |
| **CDS** | 630 | 23-kDa protein | CTVgp11 |
| **CDS** | 549 | 20-kDa protein | CTVgp10 |
| **CDS** | 360 | 13-kDa protein | CTVgp09 |
| **CDS** | 504 | 18-kDa protein | CTVgp08 |
| **CDS** | 672 | 25-kDa coat protein | CTVgp07 |
| **CDS** | 723 | 27-kDa protein | CTVgp06 |
| **CDS** | 1611 | 61-kDa protein | CTVgp05 |
| **CDS** | 1785 | 65-kDa protein | CTVgp04 |
| **CDS** | 156 | 6-kDa protein | CTVgp03 |
| **CDS** | 912 | 33-kDa protein | CTVgp02 |
| **mat_peptide** | 7818 | replicas | CTVgp01 |
| **mat_peptide** | 1475 | papain-like protease | CTVgp01 |
| **mat_peptide** | 1454 | papain-like protease | CTVgp01 |
| **CDS** | 10750 | 401-kDa viral polyprotein | CTVgp01 |
| **5'UTR** | 107 | | |

*CDS: Protein Coding Sequence.

* mat_peptide: Mature peptide.

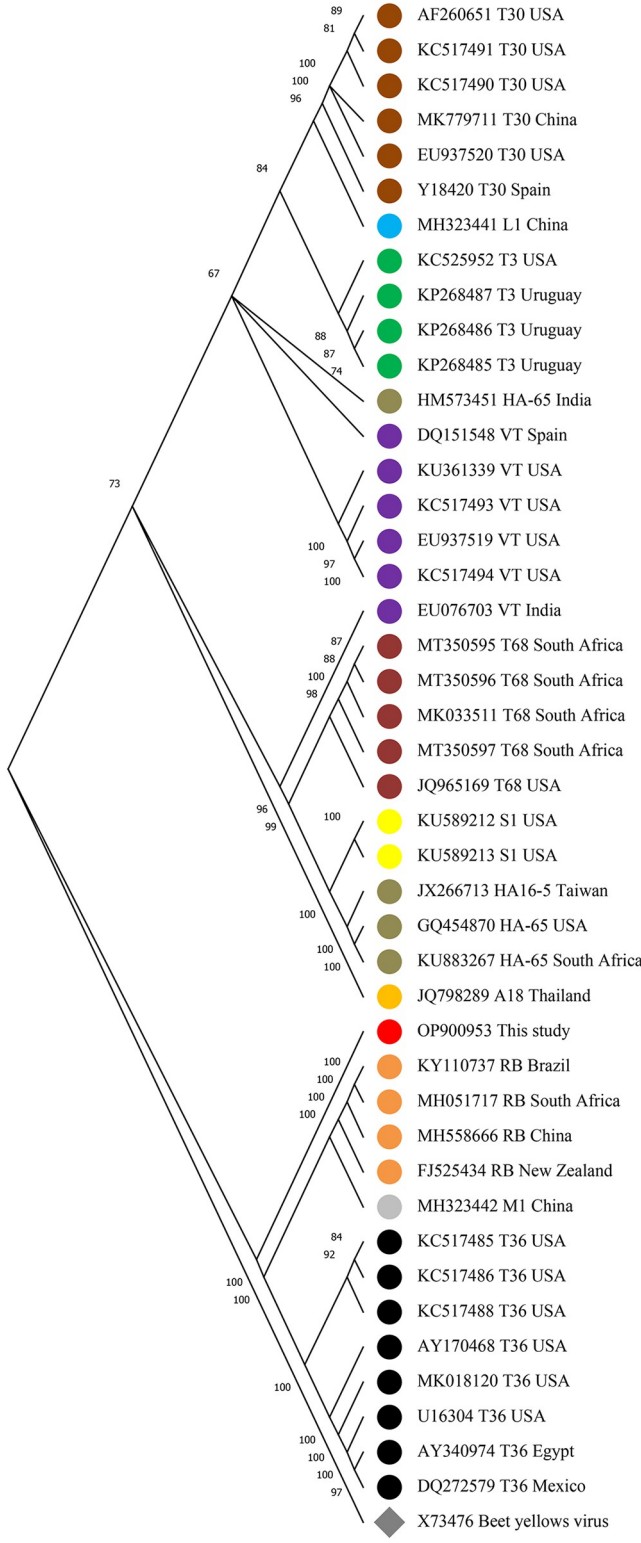

**Fig 4. Phylogenetic tree based on full-length genomes of the Sari isolate (Sari, shown with the red circle) of CTV, and the isolates from GenBank shown as accession number and country of origin.** The tree was constructed by the maximum likelihood method using MEGA 11 software. Numbers on branches are bootstrap values of 1,000 replicates. Beet yellows virus was used as an outgroup ⬤: T30, ⬤: L1, ⬤: T3, ⬤: HA-65, ⬤: VT, ⬤: T68, ⬤: S1, ⬤: HA-65, ⬤: A18, 🔴: This study, ⬤: RB, ⬤: M1, ⬤: T36, ◆: Outgroup.

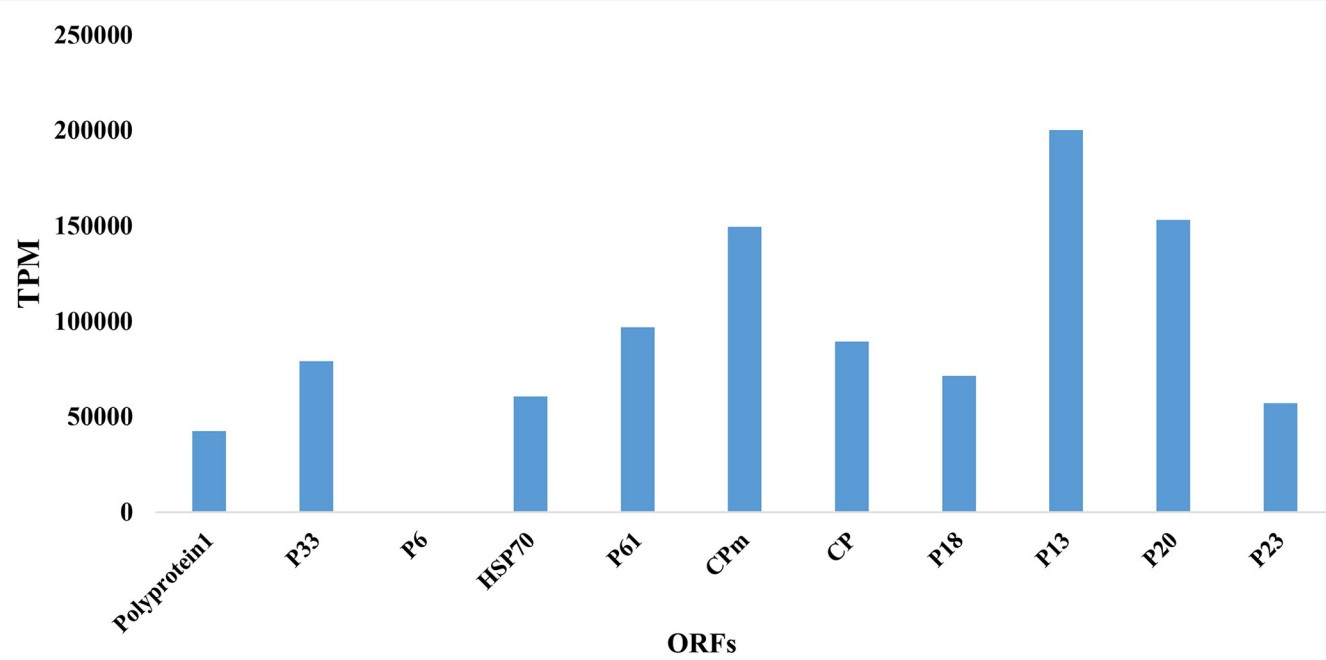

**Fig 5. Expression levels (transcript per million, TPM) of Citrus tristeza virus (CTV) Gene IDs in RNA-seq data of citrus infected with CTV.**

and P23. A quantitative comparison of TPM reads showed that the gene P13 is the most abundantly expressed transcript, followed by genes P20 and CPm. The P6 ORF showed lower TPM in this study.

The P13 protein resulted in systemic infection of calamondin was required for infection of a broader range of citrus varieties [20]. It is also reported to interact with the P20, P23 and P33 [21]. However, other functions of the P13 have not been elucidated yet and the highest expression of this gene warrants further study of its function.

The P20 protein has a key role in the suppression of RNA silencing [22] and might block the loading of CTV sRNAs into the RNA silencing complex or interfere with it through alternative mechanisms [23]. The CPm protein was the third highest-expressed gene with functions in virus movement and assembly.

These results show that CTV needs more copies of P13, P20, and CPm as they are related to the virus infection, replication and movement.

## SNV profiling on CTV draft genome

Using RNA-Seq data, this study was able to determine the SNVs on an unprecedented scale for the CTV population. For better coverage and access to all micro and macro variants of CTV in the citrus tree, we mapped the raw reads to the CTV sequence obtained from this study.

Identification of these SNVs is critical to understanding the potential of CTV variation. SNV was determined by annotating ORFs for the virus population. At least 15 sites displayed substantial differences with frequency ranging from 33% to 100% across the mapped reads when applied with a threshold of 1% for SNV detection. We filtered synonymous SNVs that did not change amino acids. The positions of SNVs in the coding regions of CTV annotated ORFs were in the polyprotein ORF (13 SNVs), P33 ORF (1 SNV) and P18 ORF (1 SNV) (Table 2). The CP genes were the most conserved and represented the same SNVs reported by

**Table 2. SNVs in the full-length genome of CTV, Sari isolate.**

| Name | Type | Reference | Allele | Count | Coverage | Frequency |
|---|---|---|---|---|---|---|
| Polyprotein Glu236Asp (Papain-like protease) | SNV | A | T | 3 | 3 | 100 |
| Polyprotein Ala242Val (Papain-like protease) | SNV | C | T | 3 | 3 | 100 |
| Polyprotein Ile246Thr (Papain-like protease) | SNV | T | C | 3 | 3 | 100 |
| Polyprotein Arg248Pro (Papain-like protease) | SNV | G | C | 3 | 3 | 100 |
| Polyprotein Ile252Leu (Papain-like protease) | MNV | GA | CC | 3 | 3 | 100 |
| Polyprotein Leu255Arg (Papain-like protease) | MNV | TC | GA | 3 | 3 | 100 |
| Polyprotein Ala313Val (Papain-like protease) | SNV | C | T | 3 | 3 | 100 |
| Polyprotein Tyr323His (Papain-like protease) | SNV | T | C | 3 | 3 | 100 |
| Polyprotein Arg327Trp (Papain-like protease) | SNV | A | T | 4 | 4 | 100 |
| Polyprotein Thr2748Ala (Replicase) | SNV | A | G | 3 | 3 | 100 |
| Polyprotein Asp2919Glu (Replicase) | MNV | TT | AC | 3 | 3 | 100 |
| Polyprotein Arg2931Pro (Replicase) | SNV | G | C | 3 | 3 | 100 |
| Polyprotein Val2934Ala (Replicase) | MNV | TA | CG | 3 | 3 | 100 |
| P33 Val111Leu | SNV | G | T | 2 | 3 | 66.66 |
| P18 Ala166Val | SNV | C | T | 2 | 6 | 33.33 |

previous studies [24]. Most SNVs in polyprotein were on papain-like proteins and four SNVs were observed in replication protein. Fig 6 shows the exact location of SNVs in the tertiary structure of the polyprotein which may suggest potential areas for further research into the mechanisms underlying virus-host interactions and understanding the molecular mechanisms underlying the effects of specific SNVs on the virus. Polyproteins are the biggest protein in CTV and have several functions.

The P33 protein has multiple functions. It is a unique non-conserved movement protein that also interacts with plant immunity [25]. In addition, the P33 has an important role in the ability of a CTV variant to protect the host from superinfection by a second closely related

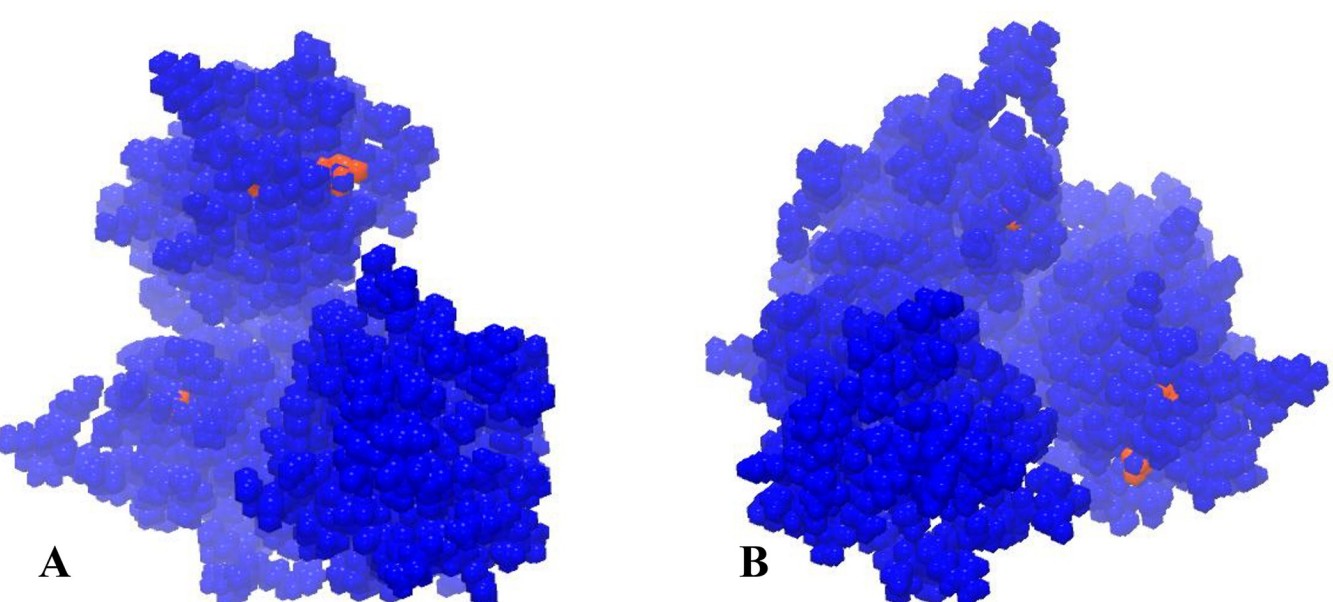

**Fig 6. The tertiary structure of polyprotein and location of SNVs (red color).** A and B are figures of polyproteins from different views.

virus variant [26]. While the CP protein of the virus is involved in virion assembly and virus translocation, P33 controls the asymmetrical accumulation of the positive- and negative-stranded RNAs during viral replication [27]. A recent study demonstrated that p33 participates in many different viral processes, and interacts with multiple protein partners such as CP, p20, and p23 [21]. The P33, P18, and P13 have been shown to expand the virus host range. For instance, several citrus genotypes can be infected with virus mutants containing deletions in those three genes [20].

The differences between strains in the different regions showed the recombination in the parental CTV genome that changed over a long time in different hosts [28]. It has been suggested that if a mixture of severe and mild strains of CTV exist in infected plant cells at different levels, the restriction of disease development can happen by a larger amount of mild viral genome, even though the other strain of the virus remains at low ratios in the viral populations [29]. Mixed viral infections of citrus trees are usually expected as they may be visited by viruliferous aphids multiple times leading to the transmission of different strains of CTV to the same tree. Consequently, it is supposed that in the numerous infected citrus trees, there are co-infections of different strains of the virus [29]. The distribution of the mild strains of CTV in a region implies good adaptation of the virus to its host [29,30]. This study does not provide direct evidence of mixed infections of different strains or variants in the sample or sequences. However, our findings do support the idea that mixed infections are possible in citrus trees.

Previous studies have shown that the CTV genotypes can be changed after passage through different hosts [31]. Studies demonstrated the existence of genotype population modification in the genotype of a single isolate after passage through two different hosts, demonstrating that the presence and dominance of population genotypes were modified by virus transmission from sweet orange to Mexican lime [31,32]. So, CTV has a variation potential in a population in a host which our results confirm previous studies and these variations can support CTV for best fit in a different situation.

## Conclusion

We sequenced and analyzed the complete genome of a CTV isolate for the first time from Iran. Phylogenetic analysis of the whole genome shows that this isolate is distinct from other isolates in the GenBank. Additionally, our study of the differential gene expression of the virus revealed that the P13 ORF was highly expressed in infected plants. Previous research has suggested that P13 plays a key role in the systemic infection and host range of the virus. Therefore, the high expression of P13 in infected plants suggests that it may be important for the virus to successfully infect and replicate within its host. The potential implications of this finding include the possibility of targeting P13 for the development of new control strategies for CTV. Our study of the potential variation of the virus in citrus trees revealed that the Sari isolate of CTV had mild variation, which could contribute to the virus's best fit in different environmental conditions. This potential variation has also been reported in previous studies [31,32], which suggests that it could be a fitness tool for the virus. Our findings add to this body of research and provide new insights into the potential for CTV to adapt to different host environments.

## Supporting information

**S1 File. Similarity matrix of full-length genomes of the Sari isolate of CTV and the isolates from GenBank.**
(CSV)

**S1 Raw images.**
(TIF)

## Author Contributions

**Conceptualization:** Abozar Ghorbani.

**Supervision:** Keramatollah Izadpanah.

**Validation:** Mohammad Mehdi Faghihi, Faezeh Falaki.

**Visualization:** Faezeh Falaki.

**Writing – original draft:** Abozar Ghorbani.

**Writing – review & editing:** Mohammad Mehdi Faghihi, Keramatollah Izadpanah.

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
