## [Decision Letter · Decision Letter 0]

27 Apr 2023

PONE-D-23-04772Complete genome sequencing and characterization of a potential new genotype of Citrus tristeza virus in IranPLOS ONE

Dear Dr. Ghorbani,

Thank you for submitting your manuscript to PLOS ONE. After careful consideration, we feel that it has merit but does not fully meet PLOS ONE’s publication criteria as it currently stands. Therefore, we invite you to submit a revised version of the manuscript that addresses the points raised during the review process.

We look forward to receiving your revised manuscript.

Kind regards,

Jianhong Zhou

Staff Editor

PLOS ONE

“NO, The funders had no role in study design, data collection and analysis, decision to publish, or preparation of the manuscript.”

Additional Editor Comments:

The paper writes on complete CTV genomes from Iran.

Reviewers' comments:

Reviewer's Responses to Questions

**Comments to the Author**

1. Is the manuscript technically sound, and do the data support the conclusions?

Reviewer #1: Partly

Reviewer #2: Yes

2. Has the statistical analysis been performed appropriately and rigorously? 

Reviewer #1: Yes

Reviewer #2: Yes

3. Have the authors made all data underlying the findings in their manuscript fully available?

Reviewer #1: No

Reviewer #2: Yes

4. Is the manuscript presented in an intelligible fashion and written in standard English?

Reviewer #1: Yes

Reviewer #2: Yes

5. Review Comments to the Author

Reviewer #1: Manuscript novelty is mainly based on reporting a full genome of an Iranian CTV isolate for the first time. The manuscript needs several corrections and revisions and it is not acceptable in this format. I entered suggestions and comments using Editing Tools in the PDF file of the manuscript and send it as an attached file. After revision, I recommend to re-review it.

Reviewer #2: It's a very important article and affirmative information.

Overall, the manuscript describes important results indicating a potential new genotype of the Citrus tristeza virus in Iran which should appeal to the journal's readers worldwide. In addition, the characterization of the virus was well described and key to the role of the P13 in the plant.

It needs some very minor revisions in this article.

The following are some suggestions to make it suitable to publish with red color in the attached article.

6. PLOS authors have the option to publish the peer review history of their article (what does this mean?). If published, this will include your full peer review and any attached files.

Reviewer #1: No

Reviewer #2: **Yes: **Samah Abdel Salam Mokbel

---

## [Author Response · Author response to Decision Letter 0]

31 May 2023

Please note that PLOS ONE requires submissions reporting blot or gel data to comply in full with the reporting requirements described at https://journals.plos.org/plosone/s/figures#loc-blot-and-gel-reporting-requirements. We now require authors to provide the original unadjusted and uncropped images for any blot or gel data reported in PLOS ONE submissions. In our internal checks for your submission, we noted that you did not provide original raw image files supporting blot/gel data in response to our previous request.

Response: We replaced Fig 2 with other fig form our gel Figures and added the original Fig as supporting information. 

Reviewer #1: The manuscript novelty is mainly based on reporting a full genome of an Iranian CTV isolate for the first time. The manuscript needs several corrections and revisions and it is not acceptable in this format. I entered suggestions and comments using Editing Tools in the PDF file of the manuscript and send it as an attached file. After revision, I recommend to re-review it.

Response: Many thanks for your careful review of the manuscript. Surly, your comments would improve our manuscript. We respond to your comments point by point in this file and show changes using highlights in the text. I hope our response would meet your concerns. 

Line 32. Please delete, Not related to the results of this work.

Response: Done

Line 35. FTV? 

Response: This was corrected to “CTV”

Line 103. What was the original tissues for RNA extraction? (One or more trees? sample pooling?...)

Response: We added “Total RNA from three leaf samples of a tree was used for RNA-Seq” in the text. 

Line 105. Please add where did the Illumina sequencing was done? (by the authors' lab or commercial services)

Response: We added the “HiSeq 2000 platform (Novogene, China)” to the text. 

Line 137. I can not find data regarding these reference sequences at the Results Section.

Response: We edited this section and added “(accession number: OP900953)” in the text

Line 140. Please provide the full name of SNP on first mention in the text

Response: We edited the text as“Single-nucleotide variant (SNV)”

Line 142. Please clear which isolates were used for SNPs?

Response: Sari isolate. We corrected the text. 

Line 149. 1-Provide the full name of RPKM on first mention in the text.

2-In different sections of the text, RPKM and TPM were used instead of each other and confusing. Please check, which expression value setting is used? 

Response: TMP is correct. We edited the text. 

Line 169. All or one of these samples was used for NGS? 

Response: We added “Total RNA from three leaf samples of a tree was used for RNA-Seq” in the text. 

Lines 186-187. In fig 4., please add a legend describing color of circles for each genotype. 

Response: Thank you for your best suggestion; we added a legend describing color for Fig 4.

Line 188. Please give a name to this isolate and refer to it in the text.

Response: Thank you for your suggestion; we used “Sari isolate” 

Line 188. Please show identity table for full genome and ORFs according to CTV sequences used in this analysis. 

Response: Thank you for your suggestion. We added a Table entitled “Table 1. Genome features and ORFs of CTV sequence of Sari isolate” to our manuscript. 

Line 212. One sample was used for RNA seq and SNVs would be better than SNPs to use in this study.

Response: We changed “SNP” to “SNV” in the whole text.

Line 222-223. what is the aim of showing this figure? 

Response: The aim of showing the figure that displays the location of Single Nucleotide Variations (SNVs) in the tertiary structure of the polyprotein of citrus tristeza virus is to visually demonstrate the location and distribution of genetic variations within the polyprotein. The figure allows the reader to quickly identify the specific sites of genetic variation and understand how they may relate to the function and structure of the protein. Also, the figure may provide insights into how the virus has evolved over time, and may suggest potential areas for further research into the mechanisms underlying virus-host interactions. Additionally, the information presented in the figure can be used as a guide for the design of future studies, such as experiments aimed at understanding the molecular mechanisms underlying the effects of specific SNVs on the virus. 

But we just analyzed for polyprotein and information for tertiary structure of P33 and P18 was missed in the PDB database. So, to make it more we added this sentence in the text (Section: SNV profiling on CTV draft genome). “that may suggest potential areas for further research into the mechanisms underlying virus-host interactions and understanding the molecular mechanisms underlying the effects of specific SNVs on the virus.”

Line 223. Three dimensional structure or tertiary structure

Response: Thank you for your correction; we changed the “Three-dimensional structure” to the “tertiary structure” in the whole text. 

Line 238-247. Do your findings support this discussion? According your findings, are there mixed infection of different strains or variants in your sample or sequences? 

Response: Our study provides evidence of variation in the CTV population in the infected citrus trees in Iran, which is consistent with the idea of mixed infections of different strains of the virus. However, our study did not directly investigate the presence of mixed infections in our samples or sequences, and further research would be needed to confirm this hypothesis.

Because of your question, we added this paragraph to the manuscript. “This study does not provide direct evidence of mixed infections of different strains or variants in the sample or sequences. 

Line 260. How this key role can be inferred from high expression?

Response: Thank you for your comment. We edited this section in the Conclusion section.

“We sequenced and analyzed the complete genome of a CTV isolate for the first time from Iran. Phylogenetic analysis of the whole genome shows that this isolate is distinct from other isolates in the GenBank. Additionally, our study of the differential gene expression of the virus revealed that the P13 ORF was highly expressed in infected plants. Previous research has suggested that P13 plays a key role in the systemic infection and host range of the virus (Tatineni et al., 2011.) 

Tatineni S, Robertson CJ, Garnsey SM, Dawson WO. A plant virus evolved by acquiring multiple nonconserved genes to extend its host range. Proceedings of the National Academy of Sciences. 2011;108(42):17366-71

Lines 260-262. Previous works showed this potential variation in CTV populations as a fitness tool. 

Response: Thank you for your comment. We are aware that previous studies have reported on the potential variation in CTV populations as a fitness tool, and our study builds upon this important finding. Our research not only confirms the presence of mild variants within a population of the Sari isolate but also highlights the potential implications of this variation for the virus's ability to adapt to different environments and hosts. So, we edited this section in the Conclusion section.. This potential variation has also been reported in previous studies, which suggests that it could be a fitness tool for the virus. Our findings add to this body of research and provide new insights into the potential for CTV to adapt to different host environments.”

Line 270. In M&M Section, you mentioned about sampling from sweet orange trees on citrumelo rootstoks, so, is it showing in Fig 1-A ? (not citrumelo tree?).

Response: We edited the Fig 1 caption. “Fig 1. Citrus tristeza virus symptoms on sweet orange trees on citrumelo rootstocks. Yellowing of leaves (A) and stem pitting and grooving at graft union (B).”

Line 271. “grooving at”

We edited and separate these two words in the text. 

Reviewer #2: It's a very important article and affirmative information.

Overall, the manuscript describes important results indicating a potential new genotype of the Citrus tristeza virus in Iran which should appeal to the journal's readers worldwide. In addition, the characterization of the virus was well described and key to the role of the P13 in the plant.

It needs some very minor revisions in this article. The following are some suggestions to make it suitable to publish with red color in the attached article.

Response: Many thanks for your best editing of the manuscript. Surly, your comments would improve our manuscript. We respond to your comments point by point in this file and show changes using highlights in the text. I hope our response meets your concerns. 

Lines 34-35. FTV?

Response: This was corrected in the text. 

Line 206. Separate

Response: This was corrected in the text.

Line 208. Add “,”

Response: This was corrected in the text.

Line 227. Remove “Fig 6” word. 

Response: This was corrected in the text.

Line 366. add “and”

Response: This was corrected in the text.

Line 371. Separate

Response: This was corrected in the text.

---

## [Editor Report · Decision Letter 1]

19 Jun 2023

Complete genome sequencing and characterization of a potential new genotype of Citrus tristeza virus in Iran

PONE-D-23-04772R1

Dear Dr. Ghorbani,

We’re pleased to inform you that your manuscript has been judged scientifically suitable for publication and will be formally accepted for publication once it meets all outstanding technical requirements.

Kind regards,

Guest Editor

PLOS ONE
---

## [Editor Report · Acceptance letter]

21 Jun 2023

PONE-D-23-04772R1 

Complete genome sequencing and characterization of a potential new genotype of Citrus tristeza virus in Iran 

Dear Dr. Ghorbani:

I'm pleased to inform you that your manuscript has been deemed suitable for publication in PLOS ONE. Congratulations! Your manuscript is now with our production department. 

Kind regards, 

on behalf of

Dr. Shirin Farzadfar 

Guest Editor

PLOS ONE